# Salivary Cytokines in Children with Nephrotic Syndrome versus Healthy Children: A Comparative Study

**DOI:** 10.3390/jcm9092691

**Published:** 2020-08-20

**Authors:** David Polak, Yael Borovitz, Dana Clyman-Levy, Yehuda Klein, Nathalie Bernfeld, Miriam Davidovits, Esti Davidovich

**Affiliations:** 1Department of Periodontics, Hadassah School of Dental Medicine, Hebrew University, Jerusalem 91905, Israel; polak@mail.huji.ac.il (D.P.);; 2Institute of Nephrology, Schneider Children’s Medical Center of Israel, Petach Tikva, and Sackler School of Medicine, Tel Aviv University, Tel Aviv 6997801, Israel; yaelbo2@clalit.org.il (Y.B.); Mdavidovits@clalit.org.il (M.D.); 3Department of Pediatric Dentistry, Hadassah School of Dental Medicine, Hebrew University, Jerusalem 91905, Israel; danadlevy@gmail.com (D.C.-L.); nbernfeld@gmail.com (N.B.)

**Keywords:** saliva, nephrotic syndrome, diagnostics, pediatric patients, cytokines, inflammation

## Abstract

Background: The aims of this study were to compare salivary cytokines and total protein between children with nephrotic syndrome (NS) and healthy children, and to examine whether saliva parameters can differentiate between steroid sensitivity and resistance and between disease remission and relapse. Methods: Twenty-seven children with nephrotic syndrome were classified according to steroid sensitivity and resistance, and disease remission and relapse. Twenty healthy children served as controls. Whole saliva samples were collected from all the participants. Urine and blood tests done on the same day as the saliva collection were recorded. Salivary total protein was quantified using bicinchoninic acid and IFNγ, IL-4, IL-8, IL-6, and IL1β levels using ELISA. Results: The mean ages of the nephrotic syndrome and control groups were 11.3 ± 2.4 and 9 ± 4.2, respectively. Compared to the control group, for the nephrotic syndrome group, total salivary protein was significantly lower, as were the levels of all the cytokines examined except IFNγ. Statistically significant differences were not found in any of the salivary markers examined between the children with nephrotic syndrome who were treatment sensitive (*n* = 19) and resistant (*n* = 8). Protein and IL-8 salivary levels were lower in the active (*n* = 7) than in the remission (*n* = 20) group. Conclusions: Salivary parameters distinguished children with nephrotic syndrome in relapse from healthy children. This may be due to decreased salivary protein excretion, which reflects decreased plasma levels, consequent to proteinuria. Accordingly, salivary markers may be developed as a diagnostic or screening tool for NS activity.

## 1. Background

Nephrotic syndrome (NS) is the most common glomerular disease in children and is manifested by massive proteinuria. The consequent decrease in circulating albumin level leads to severe reduction in oncotic pressure and generalized edema [1,2]. The proteinuria in NS stems from loss of size and charge selectivity of the glomerular basement membrane, due to activated T-lymphocytes that are mediated by circulating factors. This impairs the function of the glomerular filtration barrier [2] and results in secretion of proteins into the urine [1,2]. Urinary loss of protein causes other complications of NS, such as infections, hyperlipidemia, and hypercoagulability [1,2].

Minimal change nephrotic syndrome (MCNS) is the most common type of NS in childhood. MCNS responds well to steroid treatment, and the prognosis of long-term kidney function is good. Yet, MCNS has a high relapse rate (70–80%). The protocol for treatment of the first episode generally includes 4–6 weeks of high dose steroids, 60 mg/m^2^ daily; and then 40 mg/m^2^ on alternate days for an additional four weeks followed by gradually tapering down [3]. Most cases of MCNS remit (negative protein on urinalysis for three consecutive days [3]) within four weeks’ treatment. Non-responsive cases (about 7–8%) are defined as steroid-resistant NS (SRNS) and include histopathological types of kidney diseases other than MCNS [3].

It is unclear whether the capillary permeability in NS is limited to the kidney or affects other capillary systems such as salivary glands, which secrete oral fluids. The complex proteome of these oral fluids comprises 2340 identified proteins [4]. These proteins are derived from salivary gland secretions, plasma filtrates, cellular and bacterial derivatives, bronchial and gastrointestinal fluids, and others. Interest has grown in investigating possible applications of oral fluids in the diagnosis of systemic as well as oral diseases. Indeed, previous studies detected biomarkers in oral fluids for lung [5], breast [6] and oral cancers [7], diabetes [8], and autoimmune disorders [9].

Following from the above, salivary protein secretion in NS may reflect changes in systemic capillary permeability and changes in plasma protein levels. Thus, the current study aimed to evaluate salivary total protein and cytokine secretion in children with NS compared to healthy controls. In addition, we examined differences in salivary markers between (steroid-sensitive nephrotic syndrome) SSNS and SRNS, and also between those in remission versus the active state.

## 2. Methods

### 2.1. Study Population

Children with NS were recruited at the Institute of Pediatric Nephrology at Schneider Children’s Medical Center, Petach Tikva, Israel. The study was approved by the local Helsinki committee (Rabin Medical Center) and the Israel Ministry of Health (0466–12-RMC). The participants were categorized according to their treatment response (SSNS and SRNS) and according to whether their disease was in remittance or in the active state. During a routine visit at the clinic, the children were asked to spit into a collection tube for 5 min (unstimulated saliva). When urine and blood tests were done on the same day as the saliva collection, these were recorded.

Healthy children without a history of impaired renal function or proteinuria served as a control group and were recruited at the Department of Pediatric Dentistry at the Hebrew University Hadassah School of Dental Medicine in Jerusalem, Israel.

The study was approved by the Institutional Review Board for Research on Human Subjects (0136-16-RMC).

### 2.2. Saliva Collection

Saliva was collected in a quiet room between 8 a.m. and 12:30 p.m. The participants refrained from eating, brushing their teeth, or rinsing with mouthwash for at least 1 h before spitting. The collected saliva was kept at −80 °C until analysis.

### 2.3. Total Salivary Protein Quantification

Saliva samples were diluted ×10 in double distilled water. Protein concentration was measured using the bicinchoninic acid (BCA) protein assay kit (Thermo Fisher Scientific, Waltham, MA, USA), according to the manufacturer’s instructions.

### 2.4. Salivary Cytokine Quantification

Salivary levels of human IFNγ, IL-4, IL-8, IL-6, and IL1β were measured using ELISA kits according to manufacturer instructions (R&D systems, Minneapolis, MN, USA).

### 2.5. Statistical Analysis

Differences between values of the saliva parameters in the NS and control groups were examined using the Student’s T test. Blood test values in NS group were compared to standard mean values of children of the same age and sex using the Student’s T test. Statistical significance was set at 0.05 *p* value.

## 3. Results

The study included 27 pediatric patients diagnosed with NS and 20 healthy children as controls. In the NS group, 19 were SSNS and 8 were steroid resistant SRNS. Seven of the participants with NS had the full-blown (active) disease at the time of saliva collection, and 20 were in remission with normal urinary protein excretion.

The mean ages of the healthy and NS groups were 11.3 ± 2.4 and 9 ± 4.2, respectively. The respective proportions of females were 30% (6/20) and 52% (14/27) (Table 1). No statistically significant differences were found in the mean age and gender distribution between the SSNS and the SRNS groups (Table 2). Table 3 describes the demographic and clinical characteristics of children with nephrotic syndrome according to participants with remission and relapse.

The mean salivary total protein levels were significantly lower in the NS group than in the control group, 374.9 pg/mL vs. 479.7 pg/mL (*p* < 0.05). Mean salivary protein levels were lower in both the SRNS and SSNS groups than in the control group (Figure 1, *p* < 0.05 for healthy vs. SSNS). The mean salivary protein level was lower in the relapse than the remission group (*p* < 0.05); this corresponded with the abnormal urinary protein excretion observed.

Mean levels of the inflammatory markers examined were lower in the NS group than in the control group: 1136.23 pg/mL vs. 1913.89 pg/mL for IL-8 (*p* < 0.05, Figure 2) and 124.04 pg/mL vs. 199.02 pg/mL for IL-6 (Figure 2). Stratifying the results according to remission and relapse states did not show statistically significant differences, though the mean IL-8 level was lower in the relapse than the remission group (Figure 2). Mean levels of IL1β, IFNγ, and IL-4 were similar in the healthy, SRNS, and SSNS groups (Figure 3). Likewise, the levels of these cytokines were similar in the remission and relapse groups (Figure 3).

## 4. Discussion

The current study provides evidence that oral fluids reflect systemic changes that accompany NS and highlights the potential use of saliva as a biofluid for systemic monitoring. Furthermore, the results show a unique inflammatory pattern in children with NS, characterized by low salivary IL-8. However, this pattern does not clearly differentiate between remission and relapse states.

While the main characteristic of NS is significant proteinuria and a general decrease in circulating plasma proteins, its impact on capillary permeability in various organs, such as the salivary gland, is unknown. In the current cohort of children with NS, saliva showed a similar pattern to serum, with lower protein levels, especially in the relapse state. Therefore, despite the increased permeability of the glomerular filter in NS, capillary permeability of the salivary gland appears to be unaffected. Decreased serum protein levels correlate with changes in saliva composition that are evident in other diseases [10,11] and corroborate the intimate relation between serum and saliva characteristics. Furthermore, similar correlations have been shown between saliva and systemic markers for various conditions such as diabetes [8] and autoimmune disorders [9] in adults. In children, changes in salivary markers have also been shown for various conditions, such as autism spectrum disorder [12], Down syndrome [13], pertussis [14], pneumonia [15], and asthma [16]. The reduced circulating plasma proteins may also provide a possible explanation of the reduced salivary IL-8 levels observed in the relapse NS cases, which correlates with the reduction in salivary total protein. This change is robust in IL-8 and not in other tested cytokines since IL-8 is highly expressed in saliva glands keratinocytes and thus more likely to be detected in the saliva.

Children with NS are prone to infections due to a loss in immune system factors such as immunoglobulins in the urine [17]. Furthermore, the steroidal treatment that is administered as an immune suppressive may impair general and oral health.

Saliva is most frequently studied because its collection is easy, noninvasive, and rapid to obtain without the need for specialized equipment [18]. Moreover, the saliva has a prodigious fluid source that provides many, if not most, of the same molecules found in the systemic circulation, which makes it a potentially valuable biofluid for the diagnosis [18]. Several types of inflammatory biomarkers associated with both oral diseases as well as systemic diseases have been detected in saliva [19], of which cytokines are the most investigated ones (B). The current study supports these premises and showed reduced inflammatory markers (IL-6 and IL-8) in children with NS versus healthy children. Similarly, in their cohort of children with NS, Guimarães et al. found low levels of systemic inflammatory markers including circulating B lymphocytes, and NK and NKT cells [20].

A possible etiology of NS is abnormalities in the immune system, either in the humoral or cellular arms [21]. This is based on a number of observations. One is the sensitivity of most forms of primary NS to inhibitors of T lymphocyte: mycophenolate mofetil, corticosteroids, calcineurin inhibitors, and alkylating agents. Second is the frequent occurrence of measles and malaria following remission of NS, both of which are diseases that impair cell-mediated immunity. Third is the detection of MCNS as a paraneoplastic manifestation of lymphoreticular malignancies and other Hodgkin’s disease. Furthermore, depressed cell-mediated immunity has been observed during relapses of MCNS. This has been shown in conjunction with alterations in T cell subsets and increased cell surface expression of IL-2 receptors on T cells, reflective of T cell activation [22,23]. In a transcriptome analysis of SRNS vs. SSNS, Gyung Kang et al. recently showed a significant difference in the IL-4 pathways of mononuclear cells [24]. However, the current study did not find a difference in salivary IL-4 between the two groups. This discrepancy may be explained by the participation of immune cells other than mononuclear cells in saliva cytokine secretion. Furthermore, Gyung Kang et al. found that the differences in these pathways are associated with IL-4 receptors and not with the cytokine itself.

There are some limitations to this research. This study compromises only 26 children with NS and more studies with larger group of NS children are needed and with additional cytokines examined.

## 5. Conclusions

The current study shows that saliva is a potential alternative to serum markers for NS monitoring. This salivary approach may provide a less invasive and traumatic approach to monitor the wellbeing of NS children. Detecting salivary cytokines may serve as an additional tool for the evaluation of disease activity and response to treatment. The tested inflammatory cytokines mirrored, in part, the systemic inflammatory physiology of the participating children. However, the cytokines examined were unable to differentiate between steroid resistance and steroid sensitivity, or between the relapse and remission states. On the other hand, our findings regarding whole salivary protein levels raise the possibility of differentiating between the remission and relapse states. Further studies should investigate the use of saliva as a biofluid to monitor the systemic health of persons with NS, and also in other conditions, in both pediatric and adult populations.

## Figures and Tables

**Figure 1 jcm-09-02691-f001:**
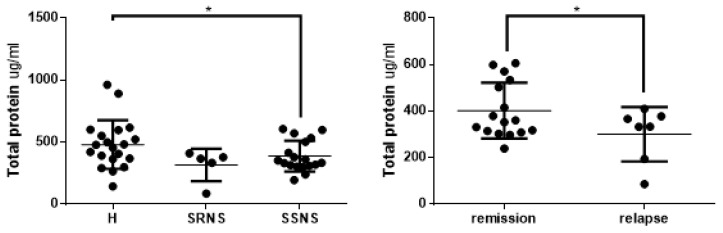
Total protein levels from the saliva of children with nephrotic syndrome and healthy controls. Samples from children with nephrotic syndrome were classified according to steroid-resistant nephrotic syndrome (SRNS) and steroid-sensitive nephrotic syndrome (SSNS); and according to the remission and relapse states. Total protein levels were quantified by a bicinchoninic acid assay. The values are presented as means and SD. * represents a statistically significant difference between the groups (*p* = 0.04 for H/SRNS/SSNS; *p* = 0.036 for remission/relapse).

**Figure 2 jcm-09-02691-f002:**
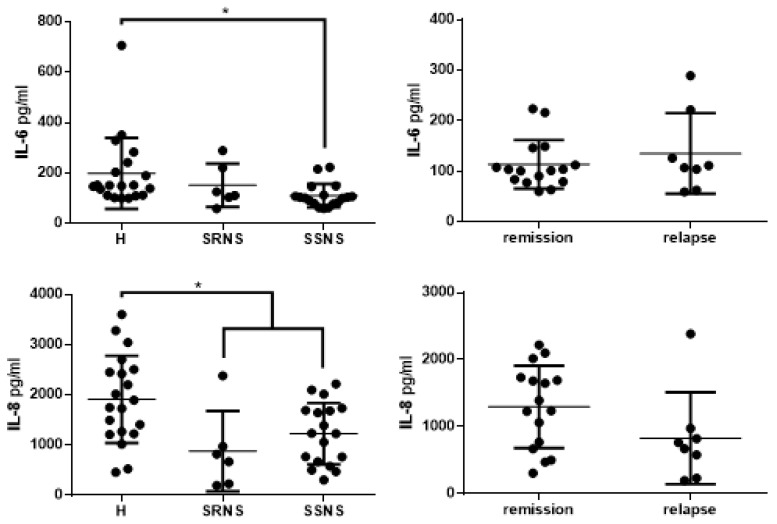
Levels of IL-6 and IL-8 from the saliva of children with nephrotic syndrome and healthy controls. Samples from children with nephrotic syndrome were classified according to steroid-resistant nephrotic syndrome (SRNS) and steroid-sensitive nephrotic syndrome (SSNS); and according to the remission and relapse states. Levels of IL-6 and IL-8 were quantified by ELISA. The values are presented as means and SD. * represents a statistically significant difference between the groups (*p* = 0.045 for IL6; *p* = 0.005 for IL8).

**Figure 3 jcm-09-02691-f003:**
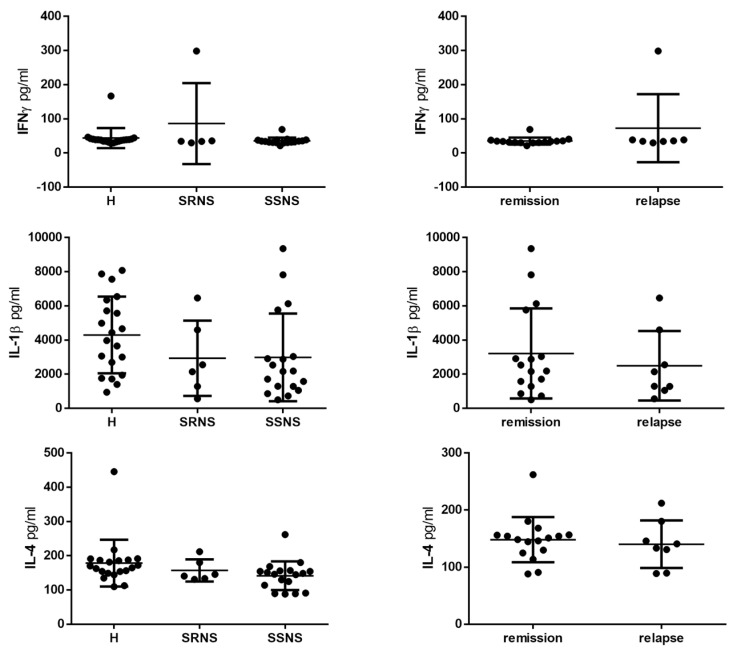
Levels of IL1β, IFNγ, and IL-4 from the saliva of children with nephrotic syndrome and healthy controls. Samples from children with nephrotic syndrome were classified according to steroid-resistant nephrotic syndrome (SRNS) and steroid-sensitive nephrotic syndrome (SSNS); and according to the remission and relapse states. Levels of IL1β, IFNγ, and IL-4 were quantified by ELISA. The values are presented as means and SD.

**Table 1 jcm-09-02691-t001:** Demographic characteristics of children with nephrotic syndrome and a control group of healthy children.

Variable	Healthy	Nephrotic Syndrome	*p* Value
Number of children	20	26	
Age	11.9 ± 2.5	9.1 ± 4.3	0.007
Gender (% of Female)	29	50	N.S.

**Table 2 jcm-09-02691-t002:** Demographic and clinical characteristics of children with nephrotic syndrome according to steroid-resistant nephrotic syndrome (SRNS) and steroid-sensitive nephrotic syndrome (SSNS).

Variable	SRNS	SSNS	*p* Value
Number of children	7	19	
Mean age, years	9.9 ± 5.5	8.8 ± 3.9	N.S
Gender (female)	4 (57%)	10 (53%)	N.S.
Positive for stick urine protein	7 (100%)	3 (15%)	<0.001
Protein/creatinine ratio	5.1 ± 4.5	2.3 ± 3.7	N.S.
Plasma albumin	3.2 ± 0.6	3.3 ± 1.2	N.S.
Plasma creatinine	0.8 ± 0.5	0.4 ± 0.1	N.S.

Data are presented as number (%) or mean ± standard deviation.

**Table 3 jcm-09-02691-t003:** Demographic and clinical characteristics of children with nephrotic syndrome according to participants with remission and relapse.

Variable	Remission	Relapse	*p* Value
Number of children	16	10	
Mean age, years	9 ± 4	9.3 ± 4.9	N.S.
Gender (female)	9 (56)	4 (40)	N.S.
Protein/creatinine ratio	0.1 ± 0	4.7 ± 3.9	0.003
Plasma albumin	4 ± 0.9	2.9 ± 0.7	0.04
Plasma creatinine	0.4 ± 0.1	0.7 ± 0.5	N.S.

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
