# Peer review of "Salivary Cytokines in Children with Nephrotic Syndrome versus Healthy Children: A Comparative Study"

_jcm, 2020, doi:10.3390/jcm9092691_

Round 1
Reviewer 1 Report
In the abstract:
Line 28-29: “Salivary parameters distinguished children with nephrotic syndrome in relapse from healthy children”. The conclusion is not aligned with the results. Please use always the same terms, active, remission or releapse?
line 30-31 “…Accordingly, salivary markers may be developed as a diagnostic or screening tool for NS activity”. NS has not been previously abbreviated
In the background:
line 50-51“It is unclear whether the capillary permeability in NS is limited to the kidney or affects other capillary systems such as salivary glands, which secrete the oral fluids.” The link supposed by the authors need to be clearer and stressed
Conclusion
“The current study shows that saliva is a potential alternative to serum markers for NS monitoring”. The authors should more explain the utility and the rationale of this method to monitoring NS and to prefer it to other methods.
Author Response
Manuscript ID: JCM-887058
Manuscript Title: Salivary cytokines in children with nephrotic syndrome versus healthy children: a comparative study.
July 2020
Dear Prof. Dr. Gianrico Spagnuolo
Guest Editor, Special Issue: "Oral Health for Special Needs, Compromised and Elderly Patients ", International Journal of Environmental Research and Public Health
We thank you for the constructive responses and remarks of the referees. The article has now been accordingly modified. Following are the specific changes that we have made:
Reviewer #1:
1. Line 28-29: “Salivary parameters distinguished children with nephrotic syndrome in relapse from healthy children”. The conclusion is not aligned with the results. Please use always the same terms, active, remission or releapse?
RESPONSE
Thank you for your comment. We distinguished children with nephrotic syndrome according to steroid sensitivity/resistance and disease remission/relapse. The above sentence relates to a relapse case.
- line 30-31 “…Accordingly, salivary markers may be developed as a diagnostic or screening tool for NS activity”. NS has not been previously abbreviated
RESPONSE:
Thank you very much for the comment. NS was abbreviated now in line 15
- line 50-51“It is unclear whether the capillary permeability in NS is limited to the kidney or affects other capillary systems such as salivary glands, which secrete the oral fluids.” The link supposed by the authors need to be clearer and stressed
RESPONSE:
This sentence stresses the fact that the pathohistological traits of NS is not fully understood. Since both kidney and salivary glands rely on capillary permeability, and it is known that in NS there is limited capillary permeability, we suggested that this may also occur in salivary glands in NS.
- Conclusion “The current study shows that saliva is a potential alternative to serum markers for NS monitoring”. The authors should more explain the utility and the rationale of this method to monitoring NS and to prefer it to other methods.
RESPONSE:
Thank you for this comment. We elaborated on this issue in the conclusion section.
We thank you again for your valuable review, which we believe has been beneficial to improve the article. We hope that the present version will be found acceptable for publication. We look forward to hearing from you as soon as possible.
Sincerely yours,
Davidovich Esti and co-authors

Reviewer 2 Report
The study by Polak et al., is well-designed and well-written. It is interesting to see that nephrotic syndrome and its steroid-resistant and steroid-sensitive types may have an effect on some salivary cytokine concentrations. I will have some minor points to address:
1) It would be good to give the LOD levels of cytokine kits. It is also needed to mention if any of the cytokine levels was left under the detection limit, and if so, how these cases were handled.
2) As authors claim that salivary IL-6 and IL-8 can be used as monitoring markers for NS, it would be good to compare these cytokines concentrations with previously published articles' results (if there is any on chilfren, if not may be on adults).
3) It would be good to mention about the periodontal status of these children as selected cytokines' concentrations are prone t periodontal disease.
4) Have authors considered to present cytokine concentrations per mg of salivary protein? If salivary total protein levels decrease in NS, it may also indicate that cytokine/mg of protein values increase in comparison to controls.
Author Response
Manuscript ID: JCM-887058
Manuscript Title: Salivary cytokines in children with nephrotic syndrome versus healthy children: a comparative study.
July 2020
Dear Prof. Dr. Gianrico Spagnuolo
Guest Editor, Special Issue: "Oral Health for Special Needs, Compromised and Elderly Patients ", International Journal of Environmental Research and Public Health
We thank you for the constructive responses and remarks of the referees. The article has now been accordingly modified. Following are the specific changes that we have made:
Reviewer #2:
- The study by Polak et al., is well-designed and well-written. It is interesting to see that nephrotic syndrome and its steroid-resistant and steroid-sensitive types may have an effect on some salivary cytokine concentrations. I will have some minor points to address:
RESPONSE
Thank you very much.
- It would be good to give the LOD levels of cytokine kits. It is also needed to mention if any of the cytokine levels was left under the detection limit, and if so, how these cases were handled.
RESPONSE
We used kits from R&D systems company which is world known for their high standard kits. The kits used in the study are not in development and as such their LOD (Level of Development) is irrelevant. No cases had results below the detection limit.
7) As authors claim that salivary IL-6 and IL-8 can be used as monitoring markers for NS, it would be good to compare these cytokines concentrations with previously published articles' results (if there is any on chilfren, if not may be on adults).
RESPONSE:
Thank you for your comment. We added a paragraph in page 6, lines 163-167. Two references were added in alphabetic order in the end of reference list.
8) It would be good to mention about the periodontal status of these children as selected cytokines' concentrations are prone t periodontal disease.
RESPONSE
This is a very important remark. Unfortunately, the IRB approval did not include a periodontal examination, and as such this information is not available for the current cohort.
9) Have authors considered to present cytokine concentrations per mg of salivary protein? If salivary total protein levels decrease in NS, it may also indicate that cytokine/mg of protein values increase in comparison to controls.
RESPONSE
This is a great suggestion. However, while such calculation may increase the comparison, it may also reflect irrelevant clinical values, we prefer to present the straight forward results and discuss them as is.
We thank you again for your valuable review, which we believe has been beneficial to improve the article. We hope that the present version will be found acceptable for publication. We look forward to hearing from you as soon as possible.
Sincerely yours,
Davidovich Esti and co-authors

Reviewer 3 Report
This manuscript by Polak et al. entitled “Salivary cytokines in children with nephrotic syndrome versus healthy children: a comparative study” presents an interesting study where the authors have employed an approach where a measure of the level of salivary parameters (total proteins) could distinguish between patients with and without nephrotic syndrome. The study is well supported by data from patients. Further, the study also suggests that the total salivary protein could be used also differentiate between the remission and the relapse states. However, careful analysis of the plot in Fig 1B suggests that majority of the data from relapse patients falls in the comparable range. Strength: The results presented here are critical as being noninvasive and economical, it has a great potential to be used as a biomarker as an alternative to serum biomarker. Shortcoming: The study is promising but the sample size has to be increased to conclude it as a potential biomarker. As far as the scope of the present manuscript is concerned, the results are interesting and it can be accepted if the minor comments below are addressed. Comments: 1. Please provide the p value and mention the number of data points as wel in the figure legends for ease of evaluation 2. The discussion should include the probable reason for lower value of IL-8 levels obtained for the relapse patients.Author Response
Manuscript ID: JCM-887058
Manuscript Title: Salivary cytokines in children with nephrotic syndrome versus healthy children: a comparative study.
July 2020
Dear Prof. Dr. Gianrico Spagnuolo
Guest Editor, Special Issue: "Oral Health for Special Needs, Compromised and Elderly Patients ", International Journal of Environmental Research and Public Health
We thank you for the constructive responses and remarks of the referees. The article has now been accordingly modified. Following are the specific changes that we have made:
Reviewer #3:
10)This manuscript by Polak et al. entitled “Salivary cytokines in children with nephrotic syndrome versus healthy children: a comparative study” presents an interesting study where the authors have employed an approach where a measure of the level of salivary parameters (total proteins) could distinguish between patients with and without nephrotic syndrome. The study is well supported by data from patients. Further, the study also suggests that the total salivary protein could be used also differentiate between the remission and the relapse states. However, careful analysis of the plot in Fig 1B suggests that majority of the data from relapse patients falls in the comparable range. Strength: The results presented here are critical as being noninvasive and economical, it has a great potential to be used as a biomarker as an alternative to serum biomarker. Shortcoming: The study is promising but the sample size has to be increased to conclude it as a potential biomarker. As far as the scope of the present manuscript is concerned, the results are interesting and it can be accepted if the minor comments below are addressed.
RESPONSE
Thank you very much
11) Please provide the p value and mention the number of data points as well in the figure legends for ease of evaluation
RESPONSE
Thank you. We added the P value to the legends of the graphs.
12 ) The discussion should include the probable reason for lower value of IL-8 levels obtained for the relapse patients.
RESPONSE
Thank you. We added this issue to the discussion.
We thank you again for your valuable review, which we believe has been beneficial to improve the article. We hope that the present version will be found acceptable for publication. We look forward to hearing from you as soon as possible.
Sincerely yours,
Davidovich Esti and co-authors

Round 2
Reviewer 1 Report
This is an interesting research about biomarker to monitoring nephrotic syndrome in pediatric population.
The main points are:
this is a small sample size and should be added in the limitations.
- the rationale is difficult to understand. Please clarify in the introduction
- the limitation should be added
- in the conclusion i suggest to better clarify how the detection of salivary cytokines can be translate in the clinical practice
Author Response
Manuscript ID: JCM-887058
Manuscript Title: Salivary cytokines in children with nephrotic syndrome versus healthy children: a comparative study.
July 2020
Dear Prof. Dr. Gianrico Spagnuolo
Guest Editor, Special Issue: "Oral Health for Special Needs, Compromised and Elderly Patients ", Journal of Clinical Medicine
We thank you for the constructive responses and remarks of the referees. The article has now been accordingly modified. Following are the specific changes that we have made:
Reviewer #1 round 2:
1. This is an interesting research about biomarker to monitoring nephrotic syndrome in pediatric population.
RESPONSE:
Thank you very much
- this is a small sample size and should be added in the limitations.
RESPONSE
This paragraph was added in the end of the discussion page 6 lines 191-193
- the rationale is difficult to understand. Please clarify in the introduction RESPONSE
As stated in the introduction in lines 50-51, the rational for this study is to examine whether the capillary permeability in NS is limited to the kidney or affects other capillary systems such as salivary glands, which secrete the oral fluids.
- the limitation should be added
RESPONSE:
This paragraph was added in the end of the discussion page 6 lines 191-193
- in the conclusion i suggest to better clarify how the detection of salivary cytokines can be translate in the clinical practice
RESPONSE:
Thank you for this comment. We elaborated on this issue in the conclusion section.
We thank you again for your valuable review, which we believe has been beneficial to improve the article. We hope that the present version will be found acceptable for publication. We look forward to hearing from you as soon as possible.
Sincerely yours,
Davidovich Esti and co-authors
